# Is Th17-Targeted Therapy Effective in Systemic Lupus Erythematosus?

**Marin Petrić** [1] and **Mislav Radić** [1,2,*]

1 Division of Rheumatology and Clinical Immunology, Department of Internal Medicine, University Hospital of Split, Center of Excellence for Systemic Sclerosis Ministry of Health Republic of Croatia, Šoltanska 1, 21000 Split, Croatia; petrinjo19@gmail.com

2 Department of Internal Medicine, School of Medicine, University of Split, Šoltanska 2, 21000 Split, Croatia

* Correspondence: mislavradic@gmail.com

**Abstract:** Systemic lupus erythematosus (SLE) is a chronic autoimmune disease with a broad spectrum of clinical manifestations. The proposed pathophysiological hypotheses of SLE are numerous, involving both innate and adaptive abnormal immune responses. SLE is characterized by the overproduction of different autoantibodies that form immune complexes, which cause damage in different organs. Current therapeutic modalities are anti-inflammatory and immunosuppressive. In the last decade, we have witnessed the development of many biologicals targeting different cytokines and other molecules. One of them is interleukin-17 (IL-17), a central cytokine of a proinflammatory process that is mediated by a group of helper T cells called Th17. Direct inhibitors of IL-17 are used in psoriatic arthritis, spondyloarthritis, and other diseases. Evidence about the therapeutic potential of Th17-targeted therapies in SLE is scarce, and probably the most promising is related to lupus nephritis. As SLE is a complex heterogeneous disease with different cytokines involved in its pathogenesis, it is highly unlikely that inhibition of only one molecule, such as IL-17, will be effective in the treatment of all clinical manifestations. Future studies should identify SLE patients that are eligible for Th17-targeted therapy.

**Keywords:** guselkumab; interleukin-17; interleukin-23; lupus nephritis; secukinumab; systemic lupus erythematosus; ustekinumab

## 1. Introduction

The immunopathogenesis of systemic lupus erythematosus (SLE) is characterized by a failure to downregulate autoreactivity, a production of autoantibodies, and an immune complex deposition causing organ damage. Systemic inflammation in SLE is maintained by the dysfunctional clearance of apoptotic debris [1]. Disease is characterized by flares and periods when disease is less active. Clinical presentations are heterogeneous, so diagnosis can be challenging and is made by using classification criteria [2]. The most common are constitutional symptoms such as fever or fatigue, musculoskeletal or skin presentations, but clinicians usually struggle with lupus nephritis, neuropsychiatric lupus, or different cytopenia [3]. The majority of patients with SLE have positive antinuclear antibodies (ANA), together with other autoantibodies [2,3]. Antimalarials, such as hydroxychloroquine, are the key treatment of all patients with SLE regardless of the disease's severity, unless when it is contraindicated. Further treatment options include different immunosuppressives, as well as intravenous immunoglobulins, when indicated. Although numerous therapeutic approaches have been tried or are under investigation, new therapy options for SLE lag behind other autoimmune diseases such as psoriasis (PsO) and rheumatoid arthritis (RA) [4].

One of SLE's characteristics is serum complement deficiency or consumption, especially in periods of flares [5]. Decreased levels of serum complement are related to the

presence of activated immune complexes in SLE, followed by the increased production of interferon (IFN) alpha and further proinflammatory cytokines, such as interleukin-17 (IL-17), IL-6, IL-2, IL-21, etc. [6,7]. These reactions are usually triggered by infection or ultraviolet radiation, but in many cases the exact cause cannot be identified. Another overwhelming circumstance in SLE is a chronic persistence of systemic inflammation, even at subclinical molecular levels. Patients with SLE have an excess of self-derived IFN inducers and a lack of negative feedback signals that downregulate the IFN response [7]. This has led to promising attempts of therapeutic blockade of type I IFN [8,9]. Further, B-cell activating factor (or B lymphocyte stimulator) and CD20 are well-known therapeutic targets in SLE. B-cell depleting therapy attenuates autoantibody production and decreases the number of activated B-cells that act as antigen-presenting cells, thus terminating self-sustaining systemic inflammation. Unfortunately, it did not prove generally effective in all presentations of the disease [10,11]. However, as pathogenicity in SLE is one of the most complex, the identification of an additional therapeutic target cytokine is pragmatic.

## 2. Interleukin 17 and Th17-Targeted Therapies

IL-17 is a proinflammatory cytokine with a defensive role predominantly against extracellular bacteria and fungi. It is involved in mucosal host-defense mechanisms, as well as in pathogenetic processes in some autoimmune and allergic diseases, such as psoriatic arthritis (PsA), spondyloarthritis (SpA), or others. This cytokine group consists of six different molecules named IL-17 A to F, and three different receptors (IL-17R) named A to C, responsible for the majority of their immunologic effects [12]. IL-17 is produced by helper CD4 positive T cells called Th17, innate lymphoid cells type 3, some CD8 positive T cells, γδ T cells, and some recently discovered cells, in response to IL-1β and IL-23 [13,14]. Taylor et al. described the subpopulation of neutrophils that produce IL-17 during infection, thus bridging innate and adaptive immunity [15]. The activation of a nuclear receptor retinoic acid-related orphan receptor gamma t (RORγt), mostly by IL-23, is responsible for the production of IL-17 and the majority of its effects, such as neutrophil recruitment, antimicrobial peptide production, and enhanced barrier function [14–16]. Despite their numerous pleomorphic functions, cytokines and receptors of the IL-17 family have unique molecular structures, not similar to any other known families of proteins, which makes them attractive therapeutic targets [17].

The available Th17-targeted therapy consists of monoclonal antibodies directed at IL-17, mostly A and F isotypes as the most active ones, IL-17R, and IL-23. The IL-17A antagonists that are widely used are secukinumab (SEC) and ixekizumab (IXE). Bimekizumab (BIM) neutralizes both IL-17A and IL-17F, and brodalumab (BRO) inhibits IL-17 by binding to IL-17R A to C [14]. The upstream neutralization of IL-23 can be accomplished with guselkumab (GUS), risankizumab (RIS), tildrakizumab (TIL), and ustekinumab (UST), which is IL-12/23 inhibitor. The blockade of RORγt is a reasonable therapeutic target for the development of small molecule inhibitors; however, so far, such a drug is not available [18–20]. Rheumatologists are experienced in SEC and IXE therapy, as well as GUS and UST therapy, mostly for PsA and SpA, while dermatologists use Th17-targeted biologics for the treatment of PsO. For PsO, PsA, and inflammatory bowel diseases (IBD), UST is used. Gómez-García et al. performed a meta-analysis about the efficacy and safety of Th17-targeted therapies in PsO. They showed that SEC is among the most effective short-term treatments but with increased incidences of adverse events (AE) or infections [21]. The most favorable safety profile was shown by UST [21]. According to Bilal et al. IXE has a slightly increased risk of withdrawal due to its toxicity compared with placebos in the chronic treatment of PsO [22]. The expected AE in all Th17-targeted therapies are increased incidence of infections, especially candida infections, and injection-site reactions for IXE, but overall, these biologics are well tolerated [23,24]. SEC, IXE and BRO are related to the exacerbation of occult IBD, so clinicians should screen for it when considering direct IL-17 inhibitors in therapy [25–27]. Although Th17-targeted therapy is considered immunosuppressive, thus favoring tumor growth, the exact role of IL-17 in cancer development

remains unclear, as its proinflammatory effects can contribute to the expansion of malignant cells [28]. The standard procedure before initiating Th17-targeted therapy includes screening for infectious diseases, such as latent tuberculosis or occult hepatitis B, and in high-risk patients, chest radiographs should be performed.

## 3. SLE and IL-17

It is well known that IFN alpha is one of the central cytokines in SLE pathogenesis. Numerous studies have shown that IFN alpha causes an increased production of IL-17, so IL-17 was defined as one of the most important proinflammatory cytokines in SLE [6,7].

### 3.1. Pathogenic Role of IL-17 in SLE

The overproduction of IL-23 and IL-17 in SLE was described over a decade ago, implicating a potential therapeutic option [29]. The production of autoantibodies, the formation of immune complexes, defective apoptosis, and the clearance of dead cells, as well as the loss of self-tolerance, can all be enhanced by the increased production of IL-17 [30,31]. Th17 cells accumulate in target organs, contributing to local IL-17 production and, eventually, tissue damage. Inflammation is maintained due to the impaired function of regulatory T cells (Treg) and Th17/Treg imbalance, as well as the neutrophil extracellular trap formation process [30,32]. Autoantibody production is one of the main features of SLE pathogenesis and Pfeifle et al. described several mechanisms in which IL-23 and dysregulated Th17-related immune responses precede the production of autoantibodies, and determine the onset of autoimmune disease [33]. In SLE, defective apoptosis and the clearance of dead cells lead to a cell-surface display of plasma and nuclear antigens that, in conditions of chronic inflammation, can induce the production of autoantibodies [31]. Further autoantibody production is enhanced by IL-17-mediated increased B-cell viability and activity [31]. The importance of IL-17 in the pathogenesis of SLE was demonstrated on a murine model by Amarilyo et al., where IL-17-deficient mice did not develop SLE [34]. IL-17 expression positively correlates with the levels of anti-double-stranded DNA (dsDNA) antibodies in a murine model of SLE induced by activated lymphocyte-derived DNA [35]. Pathogenic mechanisms in which IL-17 contributes to the clinical and animal models of SLE were summarized in reviews by Koga et al., and Li et al. [30,31]. The schematic presentation of the pathogenic role of IL-17 in SLE is shown Figure 1.

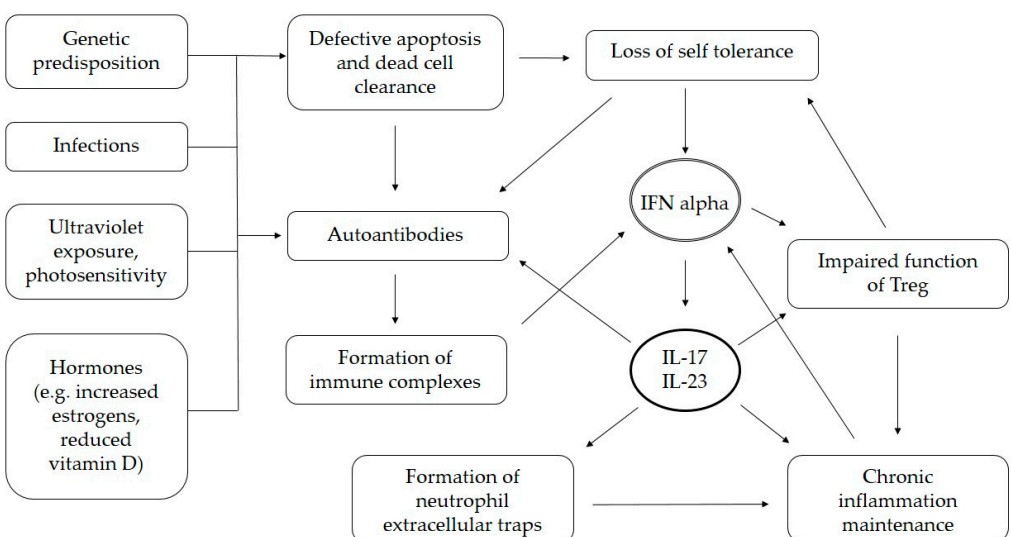

**Figure 1.** Presumed role of IL-17 in pathogenesis of SLE. IL—interleukin, IFN—interferon.

### 3.2. Human Studies of IL-17 in SLE

Numerous studies have shown exaggerated Th17 responses in patients with SLE. Studies in pediatric SLE (*n* = 40), which usually presents with an aggressive clinical course,

showed that increased serum levels of IL-17 and IL-23 are associated with higher disease activity [36]. A cytokine analysis of plasma of 60 adult SLE patients from Egypt showed significantly higher levels of IL-17 [37]. El-Akhras et al. in a recent study with 113 SLE patients and 104 controls performed a cytokine analysis and confirmed that IL-17 is significantly increased in SLE [38]. A meta-analysis of 1872 articles from 2019 about the correlation between IL-17 and SLE concluded that SLE patients have higher circulating IL-17 levels, which was influenced by ethnicity, age, disease duration, the literature quality and measurements [39]. The imbalance between Th17 and Treg in the sera of SLE patients were documented by the flow cytometry, even in patients with low disease activity [40].

The overproduction of IL-17 is commonly accompanied with increased levels of other proinflammatory cytokines, such as IL-6 or tumor necrosis factor (TNF) alpha [37]. This could explain the molecular basis of constitutional symptoms in SLE patients, such as fatigue or a prolonged low-grade fever, and the development of anemia of chronic disease. Studies about the role of IL-17 in specific organ involvement in SLE patients will be described in the next separate sections.

## 4. IL-17 Targeted Therapy and Organ Involvement in SLE

An objective problem for every clinical trial in SLE is outcome evaluation. Drug toxicities, comorbidities, and other conditions such as infections or fibromyalgia can be hardly distinguishable from SLE flares or end-organ damage. Several disease activity indexes and damage scores were designed in order to homogenize results from different studies, but the comparison of disease activity and chronic organ damage between different SLE patients still remains challenging. The Systemic Lupus Erythematosus Disease Activity Index-2K (SLEDAI-2K) and the British Isles Lupus Assessment Group-2004 (BILAG-2004) are commonly used. Recent expert validation determined Systemic Lupus Erythematosus Disease Activity Score (SLE-DAS) as an accurate and easy-to-use tool for defining the SLE clinical remission state and disease activity [41]. Future studies have to confirm the usefulness of SLE-DAS in clinical trials. In order to systematically present available evidence of Th17-targeted therapy in SLE, we performed an analysis considering common organ-specific manifestations of SLE—skin lupus, musculoskeletal manifestations, lupus nephritis (LN), neuropsychiatric lupus (NPSLE), and cytopenia.

### 4.1. Skin Lupus

SLE patients can develop numerous skin manifestations with different chronicity. The most well-known are a malar rash and discoid lesions, but annular, papulosquamous, urticarial or pemphigus-like presentations have also been described. Manifestations can be located or disseminated all over the body, with scarring or with complete skin resolution contributing to a long list of differential diagnosis [42]. In some cases, a skin biopsy and a direct immunofluorescence microscopy is needed to confirm diagnosis. Skin presentations of SLE are usually related to the presence of anti-Ro, anti-La and anti-U1-ribonucleoprotein antibodies, which mostly remain positive throughout the course of the disease [43]. Patients with skin lupus are usually photosensitive, so they are advised to use sunscreens and reduce sun exposure as much as possible. Standard local therapies for skin lupus include short-term topical corticosteroids, topical tacrolimus or pimecrolimus, or intralesional steroid applications. Further antimalarials, retinoids, or systemic immunosuppressive therapies can be used [42].

As the inhibition of IL-17 and IL-23 was found effective in PsO, similar effects were expected in skin presentations of SLE [44,45]. IL-17 stimulates keratinocytes, dendritic cells and macrophages that are located in the skin to produce chemokines, cytokines, and other proinflammatory mediators, while IL-23 causes a feedforward inflammatory response and the over-production of IL-17 [44,45]. According to Zhou et al., keratinocytes and skin plasmacytoid dendritic cells in SLE produce IFNs type I (including IFN alpha), leading to the increased recruitment of Th17 cells [6,7,46]. UST did not achieve the primary endpoint in phase III LOTUS clinical trials, and there were no significant improvements in skin

lesions versus placebos. Furthermore, a case of subacute cutaneous lupus development in a patient on UST therapy was described [47,48]. There were no other results from randomized placebo-controlled trials of IL-17 inhibitors in cutaneous manifestations of SLE available. The available evidences for SEC and BRO are limited to case reports that are not promising as treated patients developed SLE-related skin changes [49–53]. Drug-related, lupus-like skin reactions were observed in patients treated with IL-17 inhibitors [54]. However, there are two case reports from Japan of a favorable SEC treatment in patients with concomitant SLE and PsO, or PsA [55,56]. Considering IL-23 inhibitors, a case report of new-onset SLE in a patient on RIS therapy was described, as well as a case report of refractory lupus erythematosus tumidus that was responsive to TIL [57,58]. There is a lack of evidence supporting the efficacy of Th17-targeted therapy in cutaneous manifestations of SLE.

### 4.2. Musculoskeletal Manifestations

The majority of SLE patients experience joint pain at some point in time. Common musculoskeletal manifestations of SLE include non-erosive arthritis, tendinitis and tenosynovitis. In more serious cases, osteonecrosis or myopathy can occur, especially in patients who are receiving long-term or high-dose glucocorticoid therapy. Jaccoud arthropathy is a rare manifestation, which is rather specific for SLE. It usually develops due to the fibrosis of joint capsules after repeated bouts of arthritis, causing an ulnar deviation, joint deformities such as swan neck, 'boutonniere', and thumb and metacarpophalangeal joint (MCP) subluxations [59]. Erosive arthritis can also occur but usually with the presence of anti-citrullinated peptide antibodies in SLE and RA overlap known as rhupus [60]. The increased erythrocyte sedimentation rate, or levels of C reactive protein, can suggest inflammatory arthritis, but the infective process should always be previously excluded. A Swedish group of authors showed that IL-17A and IL-6 levels are elevated in the synovial fluid of SLE patients ($n$ = 17) [61]. The standard therapy includes non-steroidal anti-inflammatory drugs (NSAIDs) and other painkillers, antimalarials and immunosuppressive drugs, physical therapy, and osteoporosis treatment.

In the last few years, Th17-targeted therapy was approved for PsA and SpA after it was proven effective against active joint disease [23,62,63]. Although preclinical research showed that the exaggerated response of Th17 has a predominant role in the development of inflammatory arthritis, there are a lack of clinical trials with Th17-targeted therapy used for musculoskeletal manifestations of SLE, probably due to the mild clinical course and non-erosive arthritis [64–66]. Inflammation without swelling, tendinitis, and tenosynovitis are more prominent features in SLE, compared with synovitis and radiographical progression, which are key features in RA [66]. Considering inflammatory myopathies in SLE, we did not find clinical studies using Th17-targeted therapy for this indication, despite the fact it has been suggested as a possible therapeutic option [67,68].

### 4.3. Lupus Nephritis

LN is one of the most severe presentation of SLE associated with significant morbidity and mortality. The overproduction of immune complexes can injure renal parenchyma causing mesangial, proliferative or membranous nephritis. If disease is not treated, it results in advance sclerotic nephritis and end-stage renal disease (ESRD). There are six classes of LN according to actual histologic classification and treatment is commonly adjusted following renal biopsy [69]. Quantification of proteinuria, levels of anti-dsDNA and complement levels are used in the assessment of LN severity and activity. LN classes can overlap, even change over time, regardless of treatment. Standard therapy of active LN includes immunosuppressives, together with antimalarials, angiotensin converting enzyme inhibitors, dialysis or other supportive measures, when indicated [69].

The role of IL-17 and IL-23 in LN was studied on animal models and renal biopsy specimens of patients with LN. Although Zhang et al. showed that an aberrantly active IL-23/IL-17 axis contributes to the development of nephritis in lupus-prone mice, Schmidt et al. did not confirm a significant role of Th17 immune response in the immunopathogene-

sis of LN in murine model [70,71]. Another study showed that IL-23 receptor deficiency prevents the development of lupus nephritis in animal model [72]. The strongest evidence for Th17 related immunologic reactions in SLE was found in renal biopsy samples of patients with active LN where increased levels of IL-17 were found, especially in inflammatory infiltrates [73,74]. Levels of IL-17 are increased in sera or plasma of LN patients, and related to poor prognosis of LN [75–77]. In addition, IL-23 levels were higher in patients with positive therapeutic response, compared with non-responders [73,76]. IL-17 and IL-23 were proposed as biomarkers of disease activity, and predictors of therapeutic response in LN [73,76]. Paquissi and Abensur summed up overall effects of IL-17 in their review, considering its potential on glomerular, tubular and systemic inflammation as well as on chronic kidney damage [78]. They claim that Th17 mediated immune reactions promote repetitive tissue damage and inadequate repair, leading to increased oxidative stress, which results in accelerated apoptosis of podocytes and other renal cells, fibrosis and loss of kidney function [78].

Clinical trials SELUNE and ORCHID-LN about SEC and GUS in LN are ongoing (patients with ESRD were excluded) [79,80]. There are available case reports about effective use of SEC in LN, one in refractory LN and other in a patient with concomitant PsO [81,82]. Case series of three patients showed worsening of pre-existing LN or newly diagnosed LN in patients treated with UST [83]. Evidence of other Th17 related biologics in LN are scarce or absent, but studies showed that antimalarials or other immunosuppressants such as mycophenolic acid, when used in LN, decreased levels of IL-17 [84–86]. Additional caution is recommended when prescribing Th17-targeted therapy in patients with LN, as renal impairment and disturbed drug clearance can occur [87].

### 4.4. Neuropsychiatric, Hematological and Other Presentations of SLE

Manifestations of NPSLE can be diffuse (psychosis, delirium, headaches) or focal (cerebrovascular insults, epilepsy, demyelination). In some cases, it could be hard to distinguish between NPSLE and lupus comorbidities or therapeutic complications. Inflammation in the central nervous system (CNS) is related to presence of antibodies specific for antiphospholipid syndrome, anti-ribosomal P and anti-neuronal antibodies in serum and cerebrospinal fluid (CSF) [88]. NPSLE diagnosis is based on imaging and CSF analysis. Although NPSLE diagnosis is made after exclusion of other causes, biopsies are rarely performed. Increased serum levels of IL-17 were found in SLE patients with history of CNS disease, as well as increased levels of IL-17 in CSF in patients with NPSLE [89–91]. Th17 mediated CNS autoimmunity was described on murine models [92]. Treatment of NPSLE depends on clinical presentation and presence of antiphospholipid syndrome. It is usually symptomatic and based on prevention. Immunosuppressive treatment is indicated in acute confusional state, demyelinating disease or in the case of vasculitis. There is a lack of evidence of Th17-targeted therapy in NPSLE, possibly due to acute and unpredictable presentations, and challenging diagnosis. Caution is recommended, as low serum levels of IL-17 were described in patients with psychosis [93].

Leucopenia, thrombocytopenia, and hemolytic anemia in SLE are usually mediated by antibodies that are directed at the surface molecules of these cells and can be sensitive indicators of disease flare. An anemia can also be a consequence of inadequate iron absorption due to a chronic disease, autoimmune inflammation in bone marrow mediated by antibodies directed to erythrocyte precursors, microangiopathic hemolysis, or an adverse effect of cyclophosphamide, azathioprine, and mycophenolate therapy [94]. The standard treatment of cytopenia in SLE is directed at the underlying cause. In cases of immune mediated blood cell destruction, glucocorticoids, intravenous immunoglobulins, mycophenolate, CD20 targeted therapy, or other forms of immunosuppression are indicated. Splenectomy should be considered in resistant cases. There is weak evidence that IL-17 could contribute to the development of autoimmune hemolytic anemia (AIHA), and further studies with larger cohorts are needed [95]. Plenty of studies researched the role of Th17-associated cytokines and immune thrombocytopenia (ITP). Even though increased levels of IL-17 and IL-23 were

described in Egyptian paediatric patients with ITP, recent Chinese studies showed contradictory results, indicating that IL-17 is not strongly associated with ITP [96–99]. Genetic analysis from a rather large Chinese cohort showed that only IL-17F was associated with chronic ITP [100]. Considering SLE patients, Oke et al. showed that one fifth of patients with increased levels of IL-17 or IL-23 had thrombocytopenia [101]. In animal models, IL-17 and IL-23 stimulate proliferation and differentiation of neutrophils, and mycophenolic acid suppresses granulopoiesis in IL-17 dependent manner [102,103]. Although rare, neutropenia is stated as an expected side effect of IL-17 and IL-23 inhibitors, without increasing the risk of infection [104].

We found case reports of AIHA that were induced by SEC, and a case report of a patient with SLE, thrombocytopenia, and PsO, which were successfully treated with UST [105,106]. Other case reports presented a decline of platelets after the initiation of UST and the development of hypersplenism or thrombotic thrombocytopenic purpura [107–109]. So far, there is no available evidence of Th17-targeted therapy for immune mediated cytopenia in SLE. Clinical trials on this theme are not being performed yet, so we could not recommend Th17-targeted therapy for this indication.

Considering low-grade fever or fatigue as constitutional symptoms of SLE, as well as chronic disease anemia, it is likely that Th17-targeted therapy would alleviate these symptoms as systemic inflammation is under control [31]. If not, clinicians should search for organ damage, a different diagnosis, or another therapy option. There are case reports about the successful SEC treatment of skin lupus that is accompanied with a fever, with the resolution of hypocomplementemia, and the decrease of SLE-related autoantibodies [55,82]. The relation between Th17-targeted therapies and autoantibody production has to be elucidated. Although the levels of IL-17 correlated with levels of anti-dsDNA, IL-17 inhibition did not decrease autoantibody production in animal models [35,71]. A concise overview of indications for Th17-targeted therapy in SLE patients is shown in Table 1.

**Table 1.** Evidence of Th17-targeted therapy in SLE patients.

| Mechanism of Action | Drug | Indications | Reference |
|---|---|---|---|
| IL-17A inhibitor | Secukinumab | Skin lupus (CR), LN (CR), ongoing RCT for LN | [55,79,81,82] |
| | Ixekizumab | No data in SLE treatment | N/A |
| IL-17A and F inhibitor | Bimekizumab | No data in SLE treatment | N/A |
| IL-17 receptor inhibitor | Brodalumab | No data in SLE treatment | N/A |
| IL-23 and IL-12 inhibitor | Ustekinumab | Failed in RCT | [47] |
| IL-23 inhibitor | Guselkumab | ongoing RCT for LN | [80] |
| | Risankizumab | No data in SLE treatment | N/A |
| | Tildrakizumab | Lupus erythematosus tumidus (CR) | [58] |

CR—case report, IL—interleukin, LN—lupus nephritis, N/A—not applicable, RCT—randomized clinical trial.

## 5. Discussion

SLE is an unpredictable disease with a heterogeneous molecular background [6]. Clinicians face great challenges in the management of different SLE manifestations. Several systematic reviews defined IL-17 as a potential therapeutic target in SLE based on previous studies [30–32,67,68,78]. Koga et al. described in detail the supposed roles of IL-17 and IL-23 in SLE pathogenesis, considering the results from animal and human studies [30]. They listed sufficient evidence that Th17-driven immune reactions contribute to the development of immunopathology in patients and mice with SLE [30]. Unfortunately, it is unlikely that single-cytokine inhibition would result in a completely therapeutic response in SLE, a disease with such a large spectrum of clinical manifestations. So far, there are a lack of randomized controlled trials (RCT) on the therapeutic effects of Th17-targeted therapies in SLE. Maybe the most promising would be the therapeutic effect in LN, as clinical trials with SEC and GUS are ongoing (Table 1), and two case reports about the successful use of SEC in LN were recently published [79–82].

However, several case reports presented new-onset SLE or the deterioration of pre-existing SLE during or after Th17-targeted treatment [48–53,57,83]. Based on these few cases, SEC seems to increase the risk of lupus-like syndromes, but a similar phenomenon with an unclear molecular background was noticed in anti-TNF alpha inhibitors [110]. Probably the most disappointing of all Th17-targeted therapies in SLE were results of the phase III study of UST in SLE after promising phase II results [47,111]. To date, these are the only clinical studies of Th17-related treatment in SLE patients, while other published articles on this topic are case reports. Although the IL-17 blockade was therapeutically effective in lupus-prone mice, RCTs are needed to determine the exact role of IL-17 inhibition in human SLE. The IL-17 blockade may not be suitable for all patients and potential benefits may be limited to a subset of SLE patients whose diseases are driven by the IL-17 pathway. Biomarkers that can identify potential responders to Th17-targeted biologics are missing and this should be the focus of future research [30].

An evident weakness when making final recommendation for Th17-targeted therapy in SLE is an insufficient number of RCTs. Difficulties in SLE study designs should be considered. Maybe future clinical trials should include only subgroups of SLE patients such as patients with only skin or renal presentations, or even specific entities such as discoid lupus or LN type III. This could uniform outcomes and define precise indications for Th17-targeted therapy in specific manifestations of SLE. Another overwhelming SLE feature are disease flares, which are unpredictable with acute onset. In clinical trials, disease flares usually result in subjects' unblinding and the administration of rescue therapy. In such circumstances, case reports are valuable evidence of potential therapeutic outcomes. Our recommendations for Th17-targeted therapy in SLE are shown in Figure 2.

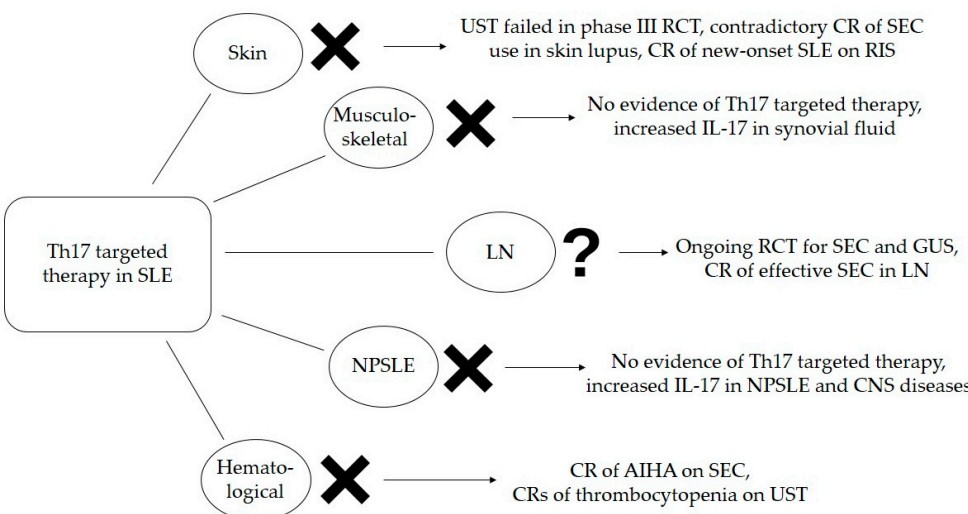

**Figure 2.** Efficacy of Th17-targeted therapy in SLE based on available evidence. AIHA—autoimmune hemolytic anemia, CNS—central nervous system, CR—case report, GUS—guselkumab, IL—interleukin, LN—lupus nephritis, NPSLE—neuropsychiatric systemic lupus erythematosus, RCT—randomized clinical trial, RIS—risankizumab, SEC—secukinumab, SLE—systemic lupus erythematosus, UST—ustekinumab.

It is unclear if increased levels of Th17-related cytokines in patients with SLE are the cause or consequence of chronic inflammation. So far, it is not defined whether IL-17 has the prominent role in early phase, chronic phase, or flares of SLE. Clinicians should not forget that many other cytokines are involved in SLE pathogenesis, apart from IL-17. The search for SLE patients that are eligible for molecular targeted therapy continues. Future studies should possibly concentrate on Th17-targeted therapy in combination with other treatment options, such as biologics, small molecule inhibitors, or well-known immunosuppressive drugs that are regularly used in the treatment of SLE today.

## 6. Conclusions

1. Strong evidence supporting the efficacy of Th17-targeted therapy in SLE is lacking.

2. Special caution is recommended when choosing Th17-targeted therapy in patients who could develop lupus-like syndromes.

3. UST was not proven effective in SLE.

4. Clinical trials with Th17-targeted therapy in LN are ongoing.

5. It is unlikely that the inhibition of only one cytokine would be effective in a heterogeneous disease such as SLE; however, in specific defined manifestations, it may be more likely.

**Author Contributions:** M.P. and M.R. contributed equally to this paper; conceptualization, M.P. and M.R.; methodology, M.P. and M.R.; writing—original draft preparation, M.P.; writing—review and editing, M.R. All authors have read and agreed to the published version of the manuscript.

**Funding:** This research received no external funding.

**Institutional Review Board Statement:** Not applicable.

**Informed Consent Statement:** Not applicable.

**Data Availability Statement:** No new data were created.

**Acknowledgments:** Authors would like to thank the personnel of Division of Rheumatology and Clinical Immunology, Department of Internal medicine, University hospital of Split, Šoltanska 1, Split, Croatia.

**Conflicts of Interest:** The authors declare no conflict of interest.

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
