# Peer review of "Is Th17-Targeted Therapy Effective in Systemic Lupus Erythematosus?"

_cimb, doi:10.3390/cimb45050275_

Round 1
Reviewer 1 Report
No surprise, agents targeting a single cytokine do not work in SLE. The authors did good review in IL-17.
My suggestions are as follows:
1."3. SLE and IL-17": This section is the core of the whole manuscript. Authors did review SLE and IL-17 from potential immunological pathways, and several studies involving IL-17 level in SLE patients.
a. Please give several paragraphs in this section.
3-1: Pathogenic role of IL-17 in SLE.
3-2: Human studies of IL-17 in SLE
b. Please provide a figure involving the possible pathogenic pathway mentioned in 3-1.
c. Please give more detailed description of studies in 3-2.
2. It is a little odd to introduce outcome evaluation in the final part of 3. SLE and IL-17. I suggest to have the "4. IL-17 and individual organ involvement in SLE. ", and 4-1,4-2... for skin, musculoskeletal... and so on. The clinical outcome parameters can be mentioned in the beginning of section 4.
Author Response
Dear Editors, dear Reviewers,
Thank you for your valuable comments on our manuscript titled “Is Th17 targeted therapy effective in Systemic Lupus Erythematosus?”. We have carefully read your remarks and addressed them as suggested. Our changes are marked in Track-changes function. Our files were processed in Microsoft Office Word 2013.
Reviewers’ comments:
Reviewer #1: No surprise, agents targeting a single cytokine do not work in SLE. The authors did good review in IL-17.
My suggestions are as follows:.
1."3. SLE and IL-17": This section is the core of the whole manuscript. Authors did review SLE and IL-17 from potential immunological pathways, and several studies involving IL-17 level in SLE patients.
- Please give several paragraphs in this section.
3-1: Pathogenic role of IL-17 in SLE.
3-2: Human studies of IL-17 in SLE
- Please provide a figure involving the possible pathogenic pathway mentioned in 3-1.
- Please give more detailed description of studies in 3-2.
Answer: In our review we focused on therapeutic potential of IL-17 inhibition in SLE patients. As You asked we included paragraphs „Pathogenic role of IL-17 in SLE“ (Page 3, Lines 12-35) and „Human studies of IL-17 in SLE“ (Page 4, Lines 2-19). We initially cited reviews by Koga et al. and Li et al. at the „Discussion“ section, they listed in details possible relations between IL-17 and SLE pathogenesis, but considering Your suggestion, we relocated them in section „Pathogenic role of IL-17 in SLE“ (Page 3, Lines 14-19, 23-26 and 30-32). We additionally explained SLE pathogenesis considering disturbances of IL-17 and included Figure 1. as You asked (Page 3, Lines 34-35). The general role of IL-17 in SLE patients was described in paragraph „Human studies of IL-17 in SLE“ and studies considering different organ involvement were described in next separate sections. Two more references (38 and 39) were included and study details were added (Page 4, Lines 3-13 and 17-19). Reference numbers were corrected subsequently.
- It is a little odd to introduce outcome evaluation in the final part of 3. SLE and IL-17. I suggest to have the "4. IL-17 and individual organ involvement in SLE. ", and 4-1,4-2... for skin, musculoskeletal... and so on. The clinical outcome parameters can be mentioned in the beginning of section 4.
Answer: As suggested, we reorganised our manuscript.
Reviewer 2 Report
The authors present a review analyzing data on a potential immunotherapy targeting Th17 cells in the treatment of SLE. The work is interesting but needs some clarifications:
Comments:
Line 11: The authors restrict in the abstract the pathogenic hypotheses to the presence of autoantibodies and the formation of immune complexes. It should be briefly stated that the pathogenetic hypotheses are numerous involving both innate and adaptive abnormal immune responses.
Line 32. the word "sometimes" should be deleted.
Line 37. The authors state that "treatment options include immunosuppressives, with exception of antimalarials and intravenous immunoglobulins when indicated." Antimalarial hydroxychloroquine is the key treatment of all patients with SLE regardless of disease severity. Therefore, the statement should be corrected.
Line 86. For clarity, the authors should specify which rheumatic conditions they are referring to regarding the efficacy and safety of SEC, UST, and IXE therapy.
Line 97. Tuberculosis should be changed to "latent tuberculosis" and hepatitis B to "occult hepatitis B." Hepatitis C is not commonly required in screening for biological therapy.
Line 150. references (40) and (41) should be inserted immediately after the efficacy of anti-IL17 and anti-IL-23 in PsO is stated. The rationale for the use of such anti-interleukin antibodies in cutaneous lupus should be better described for clarity, with specific reference to the pathogenesis of skin SLE
Line 165. The authors state that "There is a lack of strong evidence to support the efficacy of Th17 therapy in cutaneous SLE." The word "strong" should be deleted.
Line 227. The authors state that "Additional caution is recommended when prescribing Th17-targeted therapy in patients with LN, as renal insufficiency and alterations in drug clearance may occur". This statement needs a literature reference.
In Table 1, the word "indications" should be removed because biologics capable of blocking IL-17 have never been approved for any condition associated with SLE
In Figure 1, the word "recommendations" should be removed. In fact, these are only suggestions for future studies based on still partial or lacking data.
English need minor editing
Author Response
Dear Editors, dear Reviewers,
Thank you for your valuable comments on our manuscript titled “Is Th17 targeted therapy effective in Systemic Lupus Erythematosus?”. We have carefully read your remarks and addressed them as suggested. Our changes are marked in Track-changes function. Our files were processed in Microsoft Office Word 2013.
Reviewers’ comments:
Reviewer #2: The authors present a review analyzing data on a potential immunotherapy targeting Th17 cells in the treatment of SLE. The work is interesting but needs some clarifications:
Comments:
(1) Line 11: The authors restrict in the abstract the pathogenic hypotheses to the presence of autoantibodies and the formation of immune complexes. It should be briefly stated that the pathogenetic hypotheses are numerous involving both innate and adaptive abnormal immune responses.
Answer: We corrected manuscript as You asked (Page 1, Lines 11-12).
(2) Line 32. the word "sometimes" should be deleted.
Answer: As suggested, we deleted it.
(3) Line 37. The authors state that "treatment options include immunosuppressives, with exception of antimalarials and intravenous immunoglobulins when indicated." Antimalarial hydroxychloroquine is the key treatment of all patients with SLE regardless of disease severity. Therefore, the statement should be corrected.
Answer: We corrected it, as You asked (Page 1, Lines 37-40).
(4) Line 86. For clarity, the authors should specify which rheumatic conditions they are referring to regarding the efficacy and safety of SEC, UST, and IXE therapy.
Answer: Meta-analyses by Gómez-García et al. and Bilal et al. were conducted on patients with psoriasis, but in this part of our manuscript we were focused on safety of SEC, UST and IXE. As You asked we added it in the manuscript (Page 2, Lines 46 and 49).
(5) Line 97. Tuberculosis should be changed to "latent tuberculosis" and hepatitis B to "occult hepatitis B." Hepatitis C is not commonly required in screening for biological therapy.
Answer: In our country, screening for occult hepatitis C is part of a standard procedure before initiation of biologic disease modifying antirheumatic drugs. We corrected it in the manuscript as You suggested.
(6) Line 150. references (40) and (41) should be inserted immediately after the efficacy of anti-IL17 and anti-IL-23 in PsO is stated. The rationale for the use of such anti-interleukin antibodies in cutaneous lupus should be better described for clarity, with specific reference to the pathogenesis of skin SLE
Answer: We corrected as You asked. Additional explanation about pathogenesis of skin SLE and reference (46) were included (Page 5, Lines 2 and 5-7).
(7) Line 165. The authors state that "There is a lack of strong evidence to support the efficacy of Th17 therapy in cutaneous SLE." The word "strong" should be deleted.
Answer: We made correction as suggested.
(8) Line 227. The authors state that "Additional caution is recommended when prescribing Th17-targeted therapy in patients with LN, as renal insufficiency and alterations in drug clearance may occur". This statement needs a literature reference.
Answer: New reference was included (88) (Page 6, Line 32).
(9) In Table 1, the word "indications" should be removed because biologics capable of blocking IL-17 have never been approved for any condition associated with SLE
Answer: We made correction as You suggested (Page 7, Line 36).
(10) In Figure 1, the word "recommendations" should be removed. In fact, these are only suggestions for future studies based on still partial or lacking data.
Answer: As suggested, we made correction (Page 8, Line 36).
Reviewer 3 Report
This review is interesting, well structured, reads with interest, gives a fairly complete and objective presentation of the potential role of IL-17 blockade in the treatment of various manifestations of SLE. From the review, however, it remains unclear the possible (conceivable) benefits of IL-17 blockade over other immunosuppressive treatments.
Author Response
Dear Editors, dear Reviewers,
Thank you for your valuable comments on our manuscript titled “Is Th17 targeted therapy effective in Systemic Lupus Erythematosus?”. We have carefully read your remarks and addressed them as suggested. Our changes are marked in Track-changes function. Our files were processed in Microsoft Office Word 2013.
Reviewers’ comments:
Reviewer #3: This review is interesting, well structured, reads with interest, gives a fairly complete and objective presentation of the potential role of IL-17 blockade in the treatment of various manifestations of SLE. From the review, however, it remains unclear the possible (conceivable) benefits of IL-17 blockade over other immunosuppressive treatments.
Answer: Before we started to write down our manuscript, it seemed that IL-17 blockade is potential therapeutic option in SLE. Unfortunatelly, there is a lack of evidence supporting this hypothesis based on available evidence. Clinician should always perform personally oriented medicine especially in patients with SLE that can present with a wide spectrum of clinical manifestations. To the best of our knowledge, we can not recommend Th17 targeted therapy in SLE.
We hope that our explanations and corrections have adequately addressed your concerns and believe that this paper will be of great interest to the readership of your journal considering the clinical implications. Once again, we would like to thank you for your review and valuable suggestions.
With kind regards,
Marin Petrić, MD
Professor Mislav Radić, MD, PhD
Round 2
Reviewer 2 Report
The authors responded thoroughly to my comments and edited the manuscript appropriately.
English needs minor changes.